# Setup of Quantitative PCR for Oral *Neisseria* spp. Evaluation in Celiac Disease Diagnosis

**DOI:** 10.3390/diagnostics10010012

**Published:** 2019-12-26

**Authors:** Maria Valeria Esposito, Carmela Nardelli, Ilaria Granata, Chiara Pagliuca, Valeria D’Argenio, Ilaria Russo, Mario Rosario Guarracino, Paola Salvatore, Giovanna Del Vecchio Blanco, Carolina Ciacci, Lucia Sacchetti

**Affiliations:** 1Department of Molecular Medicine and Medical Biotechnologies, University of Naples Federico II, 80131 Naples, Italy; espositomari@ceinge.unina.it (M.V.E.); carmela.nardelli@unina.it (C.N.); chiara.pagliuca@unina.it (C.P.); paola.salvatore@unina.it (P.S.); 2Ceinge Biotecnologie Avanzate S. C. a R. L., 80131 Naples, Italy; dargenio@ceinge.unina.it; 3Task Force on Microbiome Studies, University of Naples Federico II, 80100 Naples, Italy; 4LabGTP (Laboratory of Genomics, Transcriptomics and Proteomics), Institute for High Performance Computing and Networking (ICAR), National Research Council (CNR), 80131 Naples, Italy; ilaria.granata@na.icar.cnr.it (I.G.); mario.guarracino@cnr.it (M.R.G.); 5San Raffaele Open University, 00166 Rome, Italy; 6Department of Medicine and Surgery, University of Salerno, 84084 Salerno, Italy; russoilaria3@gmail.com (I.R.); cciacci@unisa.it (C.C.); 7Department of System Medicine, University of Rome Tor Vergata, 00133 Rome, Italy; giovanna.del.vecchio.blanco@uniroma2.it

**Keywords:** celiac disease, oral microbiome, *Neisseria flavescens*, diagnostic marker, qPCR

## Abstract

Coeliac disease (CD) is a multifactorial autoimmune disorder and gut dysbiosis contributes to its pathogenesis. We previously profiled by 16S rRNA sequencing duodenal and oropharyngeal microbiomes in active CD (a-CD), gluten-free diet (GFD) patients, and controls (CO) and found significantly higher levels of *Neisseria* spp., with pro-inflammatory activities, in a-CD patients than in the other two groups. In this study, we developed a fast and simple qPCR-based method to evaluate the abundance of the oral *Neisseria* spp. and the diagnostic performances of the test in CD diagnosis. The *Neisseria* spp. abundances detected by quantitative PCR (qPCR) were: CO = 0.14, GFD = 0.15, a-CD = 2.08, showing a similar trend to those previously measured by next generation sequencing (NGS). In particular, *Neisseria* spp. values obtained by both methods were significantly higher (*p* < 0.001) in a-CD than in the other two groups GFD and CO—the latter almost overlapping. We calculated by ROC curve analysis the threshold of 1.12 ng/μL of *Neisseria* spp. to discriminate between CO+GFD and a-CD patients with 100% and 96.7% of diagnostic sensitivity and specificity, respectively. In conclusion, our data, if confirmed in other cohorts, suggest the q-PCR evaluation of oral *Neisseria* spp. could be a fast and simple method to assess CD-associated dysbiosis for diagnostic purposes.

## 1. Introduction

Coeliac disease (CD) is an autoimmune complex disorder triggered by the ingestion of gluten, which occurs in genetically predisposed individuals [1]. It causes enteropathy and also a large spectrum of extra-intestinal symptoms [1]. Because of its broad clinical spectrum of presentations, and the large age range at which symptoms can occur, the diagnosis could be delayed, thus causing the increasing of morbidity and mortality [1]. Although CD has a strong genetic component (mainly HLA-DQ2/DQ8 haplotypes), the environment plays a relevant role in its onset and development [2]. In particular, alterations of the gut microbiome have been reported among the several risk factors involved in CD pathogenesis [2,3].

In this context, we previously profiled by Next Generation Sequencing (NGS) of 16S rRNA the duodenal mucosal microbiome in active celiac disease (a-CD) and gluten-free diet (GFD) patients, and in a group of controls (CO), finding significantly higher levels of the Proteobacterium *Neisseria* spp. in a-CD patients than in the other two groups [4]. Interestingly, the culture-based microbiota analysis and mass spectrometry confirmed the greater abundance of Proteobacteria and identified *Neisseria flavescens* as the most contributing species to the abundance in a-CD patients. The a-CD-associated *N. flavescens* showed pro-inflammatory activities in vitro, suggesting its potential involvement in the CD-related inflammation [4]. Moreover, we found that the a-CD-associated *N. flavescens* affected mitochondrial respiration in CaCo-2 epithelial cells [5]. Next, we investigated the oropharyngeal microbiome in CD patients and controls to evaluate whether this niche shared microbial composition with the duodenum [6]. We found that *Neisseria* spp. was significantly increased, also at oropharyngeal level in the a-CD patients, with respect to both GFD patients and controls [6].

Taken together, our previous results suggest a potential role of the identified *N. flavescens* in the typical CD inflammatory *milieu* and highlight a continuum of the a-CD dysbiosis from mouth to duodenum. The above data prompted us to set up a fast, simple and cost-effective qPCR-based method to evaluate the abundance of the CD-associated *Neisseria* spp. in the oropharynx of a-CD and GFD patients compared to controls. This assay, together with the use of a less invasive sampling than the duodenum, could be useful in CD diagnosis and monitoring of GFD efficacy.

## 2. Materials and Methods

### 2.1. Patients Selection and Sampling

Patients enrolled in this study were recruited from the Departments of Gastroenterology of the Universities of Salerno and of Roma-Tor Vergata, and the Ambulatory of Molecular Medicine and Medical Biotechnologies at the University Federico II, of Naples, Italy, as previously described [6].

Among these, 45 individuals with the following characteristics were selected: 11 a-CD, on a gluten-containing diet with CD-like symptoms and positive for CD-specific antibodies (IgA anti-endomysium and/or anti-tissue transglutaminase), in whom CD was subsequently confirmed by mucosal villous atrophy of duodenum biopsies; 16 patients on GFD for at least 2 years, negative for CD-specific antibodies, and 18 CO, negative for CD-specific antibodies and without any sign of inflammatory disease.

All enrolled subjects did not present evident signs of oral inflammation (i.e., dental caries, bloody or sore gums) and had not taken antibiotics, proton pump inhibitors and anti-viral or corticosteroid in the two months before sampling.

All subjects were fully informed about the study and gave their written informed consent prior to samples collection; the study was carried out according to the tenets of the Helsinki Declaration and approved by the University of Naples Federico II Ethics Committee (Prot. N. 36/13, approval date: 25 March 2013).

Two oropharyngeal swabs (EswabTM Copan, Murrieta, CA, USA) from all study participants were collected by touching the back wall of the oropharynx and no other oral structures and stored in a Liquid Amies Elution Swab (Eswab) collection and transport system for microbiological assays. The swabs were immediately cooled with 10% glycerol in dry ice and stored at −80 °C for genetic and microbiological analysis.

### 2.2. Quantitative PCR (qPCR) Analysis

#### 2.2.1. Total DNA Extraction

Genomic DNA extracted as previously described [6] was used for qPCR analysis. DNA quantity and quality were further evaluated with the NanoDrop^®^ ND-1000 UV-Vis spectrophotometer (NanoDrop Technologies, Wilmington, DE, USA) and 0.8% agarose gel. Moreover, DNA quantity was assessed by Qubit dsDNA BR assay (Thermo Fisher, Waltham, MA, USA) according to the manufacture’s instructions. The average of three measurements was used for quantitative PCR analysis.

#### 2.2.2. Bacterial Strains Controls

The specificity of the PCR-based assays and primers selectivity for chosen target were verified by using genomic DNA from specific bacterial strains. DNA deriving from the Proteobacterium *N. flavescens*, isolated from an a-CD patient, was used as positive control; and DNA of *E. Coli* clinical isolate was used for assessing the PCR specificity and to detect any primer mismatch products.

#### 2.2.3. qPCR Amplification

*Neisseria*-specific oligonucleotide primers, described by Lansac at al. [7], were designed using PRIMER 3 v.4.1. software (available online: http://primer3.ut.ee, accessed on 23 December, 2019) and verified by Primer BLAST tool (available online: https://www.ncbi.nlm.nih.gov/tools/primer-blast/, accessed on 23 December, 2019). The sequences of the PCR primers used in this study were: Fw 5’-CTGGCGCGGTATGGTCGGTT-3’ and Rw 5’-GCCGACGTTGGAAGTGGTAAAG-3’.

Ten nanograms of genomic DNA were amplified with 10 μM of oligonucleotide primers pair and 10 μL of SYBR^®^ Green PCR Master Mix (Thermo Fisher, Waltham, MA, USA) in 20 μL of total volume reaction. The PCR mixtures were subjected to thermal cycling made up of 10 min at 95 °C, and then 38 cycles of 15 s at 95 °C for the denaturation step, 45 s at 63 °C for the annealing-extension step approximately 70 min for the 40-cycle PCR, using the 7900HT Fast Real-Time PCR System (Thermo Fisher, Waltham, MA, USA).

The DNA standard curve used for bacterial DNA quantification was done by using different dilution of the *N. flavescens* positive control DNA (from 0.2 up to 100 ng/μL).

### 2.3. Data Analysis and Statistics

Each sample was processed in duplicate; *N. flavescens* DNA was quantified plotting the cycle threshold (Ct) averages against the calibration curve. To assess statistical significance in differences of qPCR data between groups we used the Student’s *t*-test. The significance between groups was calculated through ANOVA and the post-hoc test Tukey’s “Honest Significant Difference”, both for qPCR and NGS data.

To evaluate the diagnostic value of the *Neisseria* spp. assay in distinguishing a-CD vs controls or GFD patients, we performed the receiver operating characteristic (ROC) curve analysis. The area under curve (AUC) was used to assess diagnostic specificity and sensitivity of *Neisseria* spp. evaluation as a putative biomarker for Celiac Disease. First, we compared the a-CD vs CO, then a-CD vs GFD and finally a-CD vs CO+GFD data. MedCalc software was used to develop the receiver operating curve (ROC). The *p* < 0.05 was considered statistically significant.

## 3. Results and Discussion

General features of the studied groups were previously reported [6]. The three study groups were analyzed by qPCR, as described under Methods, to evaluate the abundance of the a-CD-associated *Neisseria* spp. in each sample. Interestingly, data obtained by qPCR showed a similar trend to those previously observed for the *Neisseria* genus by NGS analysis. In fact, as shown in Figure 1, both the qPCR (panel A) and NGS (panel B) methods highlighted an increased abundance of *Neisseria* in the oropharyngeal samples of a-CD with respect to both CO and GFD patients. In detail, the median values of *Neisseria* spp. abundance detected by qPCR in the three groups were: CO = 0.14, GFD = 0.15 and a-CD = 2.08 (Figure 1, panel A), whereas those of the relative abundance previously measured by NGS were: CO = 0.12, GFD = 0.11, a-CD = 0.50 (Figure 1, panel B). In particular, the Neisseria values obtained by both methods in a-CD patients were significantly higher (*p* < 0.001) than in the other two groups of GFD and CO, whereas the values obtained in CO and GFD were overlapping.

In addition, in 7/11 a-CD patients of the present study both duodenal and oropharyngeal microbial profiles were investigated by NGS and a very similar microbial composition was found in the two niches from the same patients, being the obtained duodenal microbiomes also similar to those previously described in a different cohort of 20 a-CD patients [4,6].

Furthermore, the cultural microbiological analysis of the oropharyngeal microbiota performed in all subjects of the present study confirmed that *Neisseria* spp. was significantly (*p* < 0.001) more abundant in a-CD (13%) than in either GFD (7%) or Controls (8%) [6].

To evaluate the diagnostic value of the *Neisseria* spp. as candidate in distinguishing a-CD vs. CO and vs. GFD patients, we performed the receiver operating characteristic (ROC) curve analysis.

The ROC curve was constructed using the abundance distribution of *Neisseria* spp. in CO, GFD and a-CD groups obtained by qPCR experiments (Figure 2A–C). By the analysis of ROC curve, we calculated that the method at the threshold 1.12 ng/μL of *Neisseria* spp. abundance discriminated between CO + GFD and a-CD patients with diagnostic sensitivity 100% and specificity 96.8%, (area under the curve 0.989, 95% CI = 0.892–1.0; *p* < 0.0001).

Our data show that the qPCR assay here developed is an effective method to evaluate the abundance of *Neisseria* spp. in oropharyngeal swabs and that this measurement is able to discriminate between a-CD and GFD with high diagnostic sensitivity and specificity. The difference of *Neisseria* abundance between qPCR and NGS experiments is given to the nature of the techniques themselves and to the fact that, in case of measurement by NGS, the abundance is relative to the whole amount of the detected species. Bearing this in mind, in our previous study we confirmed, by culture-based analysis, that *N. flavescens* species mostly contributed to the significant higher abundance of *Neisseria* genus in a-CD vs GFD and controls [4,6]. Thus, supporting the evaluation here reported of *Neisseria* spp. by qPCR using *Neisseria flavescens*-specific primers.

Previous literature data have described perturbation of oral and gut microbiome in autoimmune disorder, such Rheumatoid arthritis, suggesting an overlap in the abundance and function of species at different body sites [8]. Further, a *Klebsiella pneumonia* (Kp-2H7) strain resistant to multiple antibiotics isolated in patients with Crohn’s disease, has shown an ability in colonizing and inducing intestinal inflammation in germ free mice [9]. In line with the aforementioned data, we previously found an increased presence of *Neisseria* strains isolated from both duodenal and oropharyngeal mucosa in a-CD patients when compared with controls [4,6]. The CD-associated *Neisseria flavescens* was able in inducing inflammation in dendritic cells, in ex-vivo duodenal mucosa explants of healthy controls and in CaCo-2 cells [4]. In the latter cellular model, the CD-associated *N. flavescens* also induced significant metabolic imbalance by decreasing mitochondrial respiration [5]. Opposite to the dysbiotic profile that we found in a-CD patients, both oral and duodenal microbiome recovered in GFD patients. In particular, CO and GFD patients showed almost overlapping levels of *Neisseria* strains, as expected being this Proteobacterium a member of the commensal flora found in the upper respiratory tract in healthy humans. This finding prompted us to speculate that the evaluation of the *Neisseria* spp. presence in oral samples, which is a less invasive sampling than duodenum, in addition to give further information on the *Neisseria* spp. putative pathogenetic role, could also be used for diagnostic purposes, in parallel to the CD-associated antibodies.

A limit of our study is the small cohort of CO and CD patients investigated for the setup of the assay. Further, we studied GFD patients for at least two years, when both antibodies and dysbiosis were recovered. To support our finding, the power of the assay in evaluating the efficacy of the gluten free diet has to be verified by monitoring a cohort of a-CD patients from the beginning of the GFD and in the following months, to assess whether the *Neisseria* level normalization was an earlier marker than the disappearance of CD-associated antibodies.

## 4. Conclusions

In conclusion, our data, if confirmed in other cohorts, suggest that the q-PCR evaluation of oral *Neisseria* spp. could be a fast and simple method for evaluating CD-associated dysbiosis for diagnostic purposes.

## Figures and Tables

**Figure 1 diagnostics-10-00012-f001:**
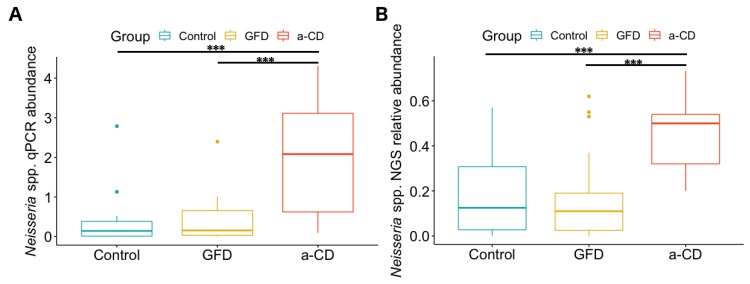
Boxplot showing the abundance distribution of *Neisseria* spp. in Control, Gluten Free Diet (GFD) and active Celiac Disease (a-CD) groups as obtained by qPCR (**A**) and by NGS analyses (**B**). Neisseria NGS relative abundance refers to a percentage with respect to the whole amount of the detected species. Boxes range from the first to third quartile and whiskers extend to the maximum and minimum values. The line inside the box is the median. Data beyond the end of the whiskers are outliers, which are plotted as points. The significance between groups was calculated through ANOVA and the post-hoc test Tukey’s “Honest Significant Difference”. Significant codes for *p*-adjusted values: *** *p* < 0.001.

**Figure 2 diagnostics-10-00012-f002:**
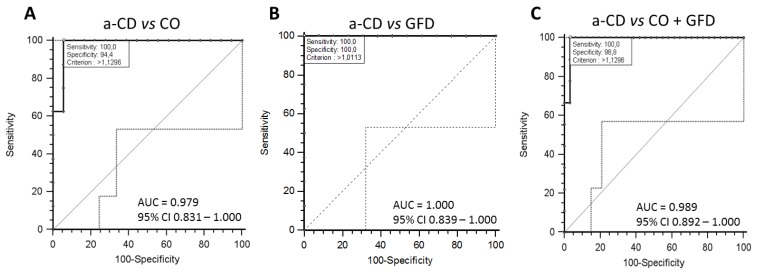
Receiver operation characteristic (ROC) curve of *Neisseria* spp. analysis using q-PCR data. First, we compared the active Celiac Disease (a-CD) vs. Control (CO) (panel **A**) and a-CD vs. Gluten Free Diet (GFD) (panel **B**), then we compared a-CD vs. CO+GFD (panel **C**). MedCalc software was used to develop the receiver operating curve (ROC). The ROC curve is indicated with bold line and open circle represents the criterion point (threshold: ng/μL). Dotted line indicates the 95% confidence interval (CI). Light line indicates the bisector. AUC, area under curve.

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
