# Peer review of "Setup of Quantitative PCR for Oral Neisseria spp. Evaluation in Celiac Disease Diagnosis"

_diagnostics, 2019, doi:10.3390/diagnostics10010012_

Round 1

Reviewer 1 Report

The authors of the manuscript " Setup of quantitative PCR for oral Neisseria spp evaluation in the celiac disease disgnosis" did a great job of identifying the species of Neisseria from the oral swab.  The authors also did the quantitative PCR and compared the quantity in Celiac disease, gluten free diet and control groups.  There are few questions to be answered:

Are these the same patients from whom the dudenal mucosal microbiome was done or are these different patients If different patients then did the authors do culture based analysis of oral samples to confirm the test.

Author Response

Reviewer 1

Comments and Suggestions for Authors

The authors of the manuscript " Setup of quantitative PCR for oral Neisseria spp evaluation in the celiac disease disgnosis" did a great job of identifying the species of Neisseria from the oral swab.  The authors also did the quantitative PCR and compared the quantity in Celiac disease, gluten free diet and control groups.  There are few questions to be answered:

Are these the same patients from whom the duodenal mucosal microbiome was done or are these different patients. If different patients then did the authors do culture based analysis of oral samples to confirm the test.

Point-by-point response to the reviewer 1 comments

We thank the reviewer for the questions that push us to add the following sentences in the paper:

In addition, in 7/11 a-CD patients of the present study both duodenal and oropharyngeal microbial profiles were investigated by NGS and a very similar microbial composition was found in the two niches from the same patients, being the obtained duodenal microbiomes also similar to those previously described in a different cohort of 20 a-CD patients (4,6).

Furthermore, the cultural microbiological analysis of the oropharyngeal microbiota performed in all subjects of the present study confirmed that Neisseria spp was significantly (p<0.001) more abundant in a-CD (13%) than in either GFD (7%) or Controls (8%) (6).”

See Results and discussion pag.4, highlighted lines 138-144.

Reviewer 2 Report

This study suggests that an evaluation of oral Neisseria spp. by q-PCR could discriminate between healthy controls+ celiacs on GFD and CD patients on normal diet.

This is an intersting and solid work and I have no major methodological concerns.

The limit of this study is the small cohort of celiacs and control subjects, but this potential weakness is clearly declared by the authors.

Author Response

Reviewer 2 Reply

Thanks.